# Increased Rates of Health Management and Health Education on Hypertension and Diabetes in Inner Mongolia, China: 10-Year Population Level Trends (2009–2018)

**DOI:** 10.3390/ijerph192013048

**Published:** 2022-10-11

**Authors:** Daxu Li, Meixuan Luo, Yu Liu, Jing Dong, Wei Geng, Xiaoliu Li, Lijun Yang, Jin Wang, Peihua Cao

**Affiliations:** 1Department of Medicine, Ordos Institute of Technology, Ordos 017000, China; 2Prevention and Health Section, Pudong New Area People’s Hospital, Shanghai 201200, China; 3Clinical Research Center, Zhujiang Hospital, Southern Medical University, Guangzhou 510280, China; 4Department of Biostatistics, School of Public Health, Southern Medical University, Guangzhou 510515, China; 5Medical Center of Diagnosis and Treatment for Cervical and Intrauterine Diseases, Obstetrics and Gynecology Hospital of Fudan University, Shanghai 200032, China; 6Department of Liver Surgery, Renji Hospital, School of Medicine, Shanghai Jiaotong University, Shanghai 200127, China; 7Department of Rehabilitation Medicine, Minhang Hospital, Fudan University, Shanghai 200032, China; 8SUMHS-SHUANG JIA Institute of Emergency Medical Rescue Technology, Shanghai University of Medicine and Health Sciences, Shanghai 201318, China

**Keywords:** chronic disease management, health education, basic public health service, multistage cluster stratified random sampling, Inner Mongolia

## Abstract

Health management and health education are two important tasks in the national basic public health service project with a wide audience, large service volume, and high accessibility. From 2009 to 2018, the Inner Mongolia Autonomous Region of China launched the basic public health service (BPHS) project comprehensively. The implementation of health management and health education was supported and instructed actively. This study aimed to document population-level trends in health management and health education on chronic diseases such as hypertension and diabetes in Inner Mongolia, China. We collected monthly and annual reports on the implementation progress of the BPHS project in Inner Mongolia, China. A two-stage random sampling method was used to investigate health management and health education for hypertension and diabetes patients. The rate of standard health management for both hypertension and diabetes has significantly increased. The blood pressure control rate and glycemic control rate have also improved. This work provides the most comprehensive evidence to date regarding the upward trends in health management and health education on chronic diseases such as hypertension and diabetes in Inner Mongolia, China.

## 1. Introduction

Chronic disease, short for chronic non-communicable disease (NCD), is one of the major health concerns, with high morbidity, disability, mortality, and high treatment costs worldwide [1,2]. Hypertension and diabetes are the two most common chronic conditions and can seriously affect life expectancy and bring significant economic burdens [3]. For hypertension, according to findings from two nationwide studies in China, one in four adults had hypertension in the period between 2012 and 2015, as well as low awareness and control rates, which indicates gaps in public health service and clinical care [4,5]. A nationally representative cross-sectional survey in 2013 in China, which consisted of 170,287 participants, found that only 37% of participants with diabetes were aware of their diagnosis, and only 32% of them were being treated [6].

The World Health Organization (WHO) Global Action Plan for the Prevention and Control of NCDs 2013–2020 recommends that a strengthened health system directed towards addressing NCDs should aim to improve the prevention, early detection, treatment, health education, and sustained management of people with NCDs [7]. As part of China’s new health care reform, initiated in 2009, the Chinese government has launched a basic public health service (BPHS) project to provide universal and equitable access to high-quality health care for all citizens [8,9,10]. It is also a necessary part of the reform of the primary-level medical and health system [11]. Chronic disease management and health education with a broader audience, large service quantity, and high accessibility are two critical tasks in the BPHS [12]. A major aim of the BPHS is to combat the increasing burden imposed by NCDs through a range of measures for the management of hypertension and diabetes, including health education, improving medication compliances, controlling risk factors such as smoking and alcohol intake, and combating obesity, in line with the recommendations by the WHO for essential packages of interventions for NCDs by primary health care (PHC) sectors [13].

A national assessment of BPHS delivery in primary health care sectors was conducted in 2010, indicating regional disparities in China. However, the study only evaluated BPHS delivery in the Eastern, Central, and Western regions in China, while the progress and challenges of BPHS in Northern China remained unknown [14]. Northern China is an underdeveloped area in China, whose Gross Domestic Product (GDP) per capita is much lower than that of Eastern and Central China. The Inner Mongolia Autonomous Region is the most important and representative province in Northern China. According to the 2020 China Population Census data, Han and Mongolians account for approximately 79% and 18% of the total population, respectively, and are the major ethnicities in the region. Previous studies showed that the prevalence of hypertension and obesity was significantly higher in Mongolians than in Han [15,16].

Although BPHS has been carried out in Inner Mongolia since 2009, there is no systematic evaluation of BPHS delivery in this region. Therefore, we selected Inner Mongolia as the study region to assess the achievements and shortfalls to implement chronic disease management and health education in underdeveloped Northern China to then better address the health disparities in China.

## 2. Data and Methods

A retrospective analysis was conducted to elaborate on implementing the chronic disease management and health education project in Inner Mongolia from 2009 to 2018. The regional-level information on the implementation progress of BPHS was collected from the monthly or annual reports from the Center for Disease Prevention and Control, the Office of Patriotic Health Campaign Committee, and other relevant health authorities in the Inner Mongolia Autonomous Region.

We used a two-stage sampling method to investigate health management and health education for hypertension and diabetes patients. The first stage was stratified random sampling for city districts or banners. A banner is an administrative unit of the Inner Mongolia Autonomous Region in China, which is equivalent to a township-level administrative division. The city districts were defined as urban areas, where local governments are located, while banners were defined as agricultural and pastoral areas or rural areas. We randomly selected 2 city districts and 7 banners. The second stage was random sampling for primary health care sectors, such as community health centers in urban areas, and township hospitals and village clinics in rural areas. Residents’ health records, health education, chronic disease management information, and other relevant data from 2009 to 2017 were collected from selected primary health care sectors. We randomly selected 70 primary health care sectors, including 21 community health centers, 18 township hospitals, and 31 village clinics. Descriptive statistical methods and line or bar graphs were used to describe the trends of health management and health education management.

## 3. Results

Health information technology (HIT) systems for BPHS are essential for the continuity and coordination of primary health care. Health data management is one of the most important roles of the HIT system, including tracking residents’ health status and treatment. Since the launch of BPHS in 2009, the Inner Mongolia Autonomous Region has made efforts to promote the electronic medical record (EMR) system for all residents, especially for key groups such as children aged 0–6, pregnant women, the elderly, and patients with chronic diseases. Along with the increasing GDP per capita in Inner Mongolia, the EMR registration rate increased from 14.36% in 2009 to 85.79% in 2018 (Figure 1). At the same time, the active EMR utilization rate also increased year by year, reaching 53.09% in 2018 (Figure 1).

Another important function of the BPHS is to deliver health education to the public. From 2009 to 2018, diverse health education activities, conducted by primary health care sectors, were freely provided for both urban and rural residents to disseminate health care knowledge. The health education activities include the provision of health education materials, such as printed and audio-video materials, propagandizing columns of health education (PCHE), and health lectures. The number of both printed and audio-video health education materials distributed and played dramatically increased from 2009 to 2018 (Table 1). More than 1000 PCHEs were made each year and frequently updated to deliver health care information and knowledge to the public (Table 1). Since 2014, the number of health lectures held was over 4000 across the whole region, with a rising number of participants from 223,300 in 2009 to 1,857,200 in 2018 (Table 1).

Apart from the autonomous region-level health management and health education information, we further systematically sampled 70 primary health care sectors, including 21 community health centers in urban areas, 18 township hospitals, and 31 village clinics in rural areas, to provide in-depth descriptions of health management on chronic diseases such as hypertension and diabetes in Inner Mongolia. The BPHS started to implement standard management for hypertension patients in 2009, including regular physical examination and health guidance on medication, diet, exercise, and psychology. At the same time, various health education activities such as health lectures and brochures on hypertension knowledge were provided regularly by the primary health care sectors. In 2010, the BPHS in Inner Mongolia began to implement a blood pressure measurement system that provides free blood pressure measurement for residents over 35 years old. After ten years of BPHS implementation, the standard management rate of hypertension stayed over 65% (Figure 2). The blood pressure control rate increased from 42.0% in 2010 to 65.4% in 2017 (Figure 2). Antihypertension medication is essential in blood pressure control. The BHPS provided 20 antihypertensive drugs in 2007, which doubled the number in 2009 (Table 2). The proportion of types of antihypertensive drugs was up to 6.83% among all social medical insurance medications (Table 2).

Similarly, the standard management rate of diabetes also increased in the past decade in Inner Mongolia. Glycemic control is one of the key means of controlling diabetes. The glycemic control rate of diabetic patients has consistently remained over 40% and showed an upward trend from 2011 to 2017 (Figure 2). In addition, the early-stage management of diabetes, including screening for diabetes, the standard management of pre-diabetes, and diabetes health education, could also effectively control diabetes. Besides, screening for the chronic complications of diabetes could improve the quality of life of diabetic patients and delay disease progression. In 2009, 343 communities conducted screening for diabetes, 314 communities managed pre-diabetic patients, and 282 communities conducted screening for chronic complications of diabetes. Those numbers showed rising trends in the past decade, increasing to 515, 500, and 384, respectively, in 2017 (Table 3).

## 4. Discussion

Since China launched the BPHS project in 2009, the government has introduced several policies to build an integrated health delivery system based on primary health care to prevent and manage chronic diseases. More and more NCD patients were diagnosed early, followed up with, and treated. In Inner Mongolia, the BPHS project has reached over 68% of hypertensive patients and 66% of diabetic patients. A cross-sectional health service interview survey across 17 provinces in Eastern, Western, and Central China in 2014 indicated a significantly better control rate for hypertension in patients ≥ 35 years under the BPHS project compared to those not under management (control rate: 76.9% vs. 68.2%) [17]. An evaluation study conducted in five provinces in China showed a positive association between BPHS implementation and diabetes control in China during the 10-year implementation from 2009 to 2019 [18]. Blood pressure monitoring and blood glucose control are key measures in the standard management of hypertension and diabetes. Better control of blood pressure and blood glucose was observed after BPHS implementation in Inner Mongolia. The increasing trends were also found in other regions of China, which showed that the reported blood pressure control rate was 50%–65% in hypertensive patients, and the blood glucose control rate was 50%–70% in diabetic patients [19]. The number of communities that provided fasting blood glucose tests and glycosylated hemoglobin (HbA1c) tests showed an increasing trend.

Besides, lifestyle management is also an effective way to control NCDs such as hypertension and diabetes [20]. Many studies on the control of hypertension and diabetes suggested the prominent roles of lifestyle factors, including smoking, alcohol drinking, and exercise, as well as socio-economic factors, such as income and education [21,22]. The China Health and Retirement Longitudinal Study (CHARLS) from 2011 to 2015 showed an acceptance rate of 70%–83% for smoking cessation, a rate of 66%–70% for alcohol quitting, and a rate of 28%–39% for body weight control in hypertensive and diabetic patients aged above 45 years old [23]. The guidelines of the BPHS project require primary health care sectors to “Conduct health education for all patients to encourage life style behavioral improvement objectives and evaluate the progress in follow-up visits, letting patients know when to visit doctors immediately”. Thus, in addition to drug therapy, more and more patients with hypertension and diabetes in Inner Mongolia accepted lifestyle intervention as well.

Health education is one of the most important programs of the BPHS [10]. Conducted by the primary health care sectors, health education is freely provided for residents to disseminate health care knowledge, especially the knowledge of NCDs such as hypertension and diabetes [24,25]. During the implementation of the BPHS in Inner Mongolia, residents’ health literacy has been improved through various forms of health education, such as the provision of health education materials, health consultations, health lectures, and follow-up visits. The increase in health education attendance of residents can also urge healthcare workers (HCWs) to strengthen their knowledge and skills to improve their professional level.

Since the BPHS project was implemented in 2009, the residents’ EMR registration rate in Inner Mongolia has increased from 14.4% in 2009 to 84.16% in 2017, and the active EMR utilization rate has also reached 47.76%. However, there are still some shortfalls in managing residents’ health records. First, there are still missing items in the residents’ health records, mainly due to incomplete and non-compliant fillings. Second, the resident health record system needs continuous improvement. A high-performance IT system would be able to capture, organize, and normalize data from many sources, maintain data securely, grant access to data selectively, and provide the computational power to rapidly analyze data [26]. Therefore, the overall quality of residents’ health records needs to be further improved.

The practical and policy implications of this study are threefold. Firstly, this study described the increasing trends of health control on hypertension and diabetes in Inner Mongolia, which reflected the effects of the BPHS. Secondly, residents’ health literacy will be improved by health education. Effective health education for residents should be further enlarged and strengthened. Research and training on how to carry out effective health education by HCWs in PHC sectors deserve consideration. Thirdly, the BPHS project could provide even more benefits if existing healthcare management processes could be optimized, such as improving the information management system.

Though our findings may not generalize to other regions in Central and Eastern China, where the socio-economy developed better compared to Northern China, the experiences may be helpful for other regions in China or developing countries with the same socio-economic characteristics. This study has several limitations. First, our study comprises a series of cross-sectional data retrieved from monthly or annual reports of health authorities, so longitudinal changes in the risk factors of NCDs at an individual level could not be evaluated. Second, our retrospective study may not be able to evaluate the relationship between the management provided by the BPHS project and hypertension and diabetes control. An experimental or quasi-experimental design is needed for further evaluation of the performance of the BPHS project. Third, logic models such as generic evaluation frameworks were not employed due to the lack of in-depth interviews and surveys. However, the regional-level monitoring data may indirectly reflect the effects of the BPHS on hypertension and diabetes management.

## 5. Conclusions

This study provides the most comprehensive evidence to date regarding the upward trends in health management and health education on chronic diseases such as hypertension and diabetes in Inner Mongolia, China. We found that the rate of standard health management for both hypertension and diabetes has significantly increased. The blood pressure control rate and glycemic control rate have also improved. Therefore, the increasing trends in health management and health education in the last decade will provide important theoretical support and practical experience for the construction of a healthy Inner Mongolia and healthy China.

## Figures and Tables

**Figure 1 ijerph-19-13048-f001:**
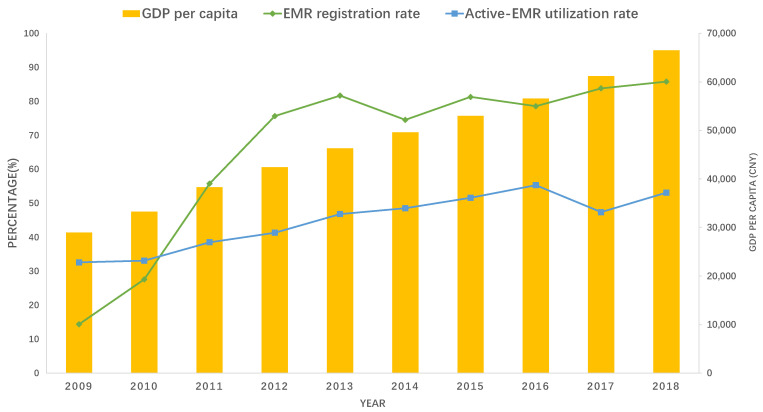
The GDP per capita, EMR registration, and active EMR utilization rates from 2009 to 2018 in Inner Mongolia, China.

**Figure 2 ijerph-19-13048-f002:**
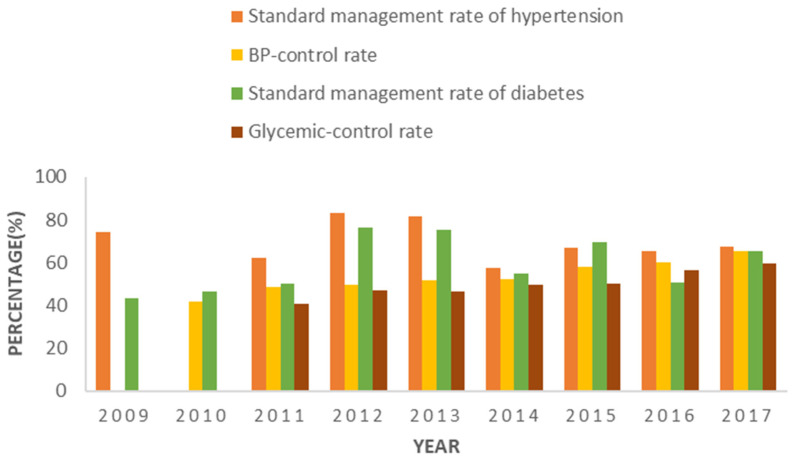
The rates of standard management for hypertension and diabetes, blood pressure control rates, and glycemic control rates from 2009 and 2017 in Inner Mongolia, China.

**Table 1 ijerph-19-13048-t001:** Health education for residents in Inner Mongolia from 2009 to 2018 (shown in ten thousands).

Year	No. of Printed Materials Distributed	No. of Audio-Visual Materials Played	No. of PCHE *	No. of Updates of PCHE *	No. of Health Lectures Held	No. of Participants in the Lectures
2009	155.18	8.46	0.17	1.07	0.65	22.33
2010	340.90	57.92	0.31	1.76	1.24	58.35
2011	293.89	56.52	0.97	2.20	2.24	80.56
2012	2088.23	78.43	1.08	4.28	2.99	132.68
2013	1515.46	64.58	0.90	3.81	3.70	185.00
2014	1182.50	40.10	1.61	5.34	4.01	143.77
2015	1463.44	58.16	1.44	6.47	5.02	223.53
2016	1671.37	69.08	1.55	7.17	4.84	249.56
2017	1378.60	60.71	1.26	5.80	4.91	141.92
2018	1446.41	62.84	1.60	7.24	4.07	185.72

*: PCHE refers to propagandizing column of health education.

**Table 2 ijerph-19-13048-t002:** The number of types and proportion of antihypertensive medications provided by the basic public health service project in Inner Mongolia, 2009–2017.

Year	Types of Antihypertension Medications	Proportion of Medications (%)
2009	10	4.67
2010	12	4.98
2011	14	5.54
2012	15	5.81
2013	15	5.86
2014	17	6.52
2015	18	6.83
2016	18	6.77
2017	20	6.82

**Table 3 ijerph-19-13048-t003:** Numbers and proportions of communities provided with early-stage management of diabetes from 2009 to 2017.

Year	Communities Conducting Diabetes Screening	Communities Managing Pre-Diabetic Patients	Communities Conducting Screening for Chronic Complications of Diabetes
2009	343 (97.72%)	314 (89.46%)	282 (80.34%)
2010	360 (98.36%)	329 (89.89%)	289 (78.96%)
2011	398 (98.51%)	365 (90.35%)	309 (76.48%)
2012	418 (98.58%)	388 (91.51%)	315 (74.29%)
2013	424 (98.15%)	408 (94.44%)	318 (73.61%)
2014	453 (97.42%)	440 (94.62%)	340 (73.12%)
2015	481 (97.96%)	465 (94.7%)	363 (73.93%)
2016	490 (97.42%)	476 (94.63%)	374 (74.35%)
2017	515 (97.53%)	500 (94.69%)	384 (72.73%)

## Data Availability

Restrictions apply to the availability of these data.

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
