# Peer review of "Increased Rates of Health Management and Health Education on Hypertension and Diabetes in Inner Mongolia, China: 10-Year Population Level Trends (2009–2018)"

_ijerph, 2022, doi:10.3390/ijerph192013048_

Round 1

Reviewer 1 Report (Previous Reviewer 1)

Dear authors, 

The paper has been improved after revision. I am satisfied with the result of the review.

This manuscript is a resubmission of an earlier submission. The following is a list of the peer review reports and author responses from that submission.

Round 1

Reviewer 1 Report

The authors state that the study aims to elaborate on implementing the chronic disease management and health education project in Inner Mongolia from 2009 to 2018. To achieve the proposed objective, the retrospective analysis was carried out on resident electronic health records registration and adoption, health education, and health management on hypertension and diabetes to analyze the trends and existing problems in the past decade. The results present ample evidence regarding the upward trends in health management and health education on chronic diseases such as hypertension and diabetes in Inner Mongolia, China. The article presents evidence on the important public health service project (BPHSP). On the other hand, the paper needs significant revision before being considered for publication.

1 - The introduction needs improvement. I suggest it be highlighted a) subject-object of study; (b) previous works that have already analyzed the object; (c) justification and originality of its approach; (d) international interest; (e) research question not yet fully answered.

3 - Why was the study conducted in Mongolia and not other regions? Can the results of this study be generalized? What is the importance of this study for the international public? Make this clear in the text.

4 - "Part of the relevant data was gained from the monthly or annual report and a particular enterprise report of the implementation progress of BPHSP from 2009 to 2018 that are regularly reported by the national basic public health service projects".

-What are these data?

- Why was the analysis from 2009 to 2018 and not another period? For example, figures 1 and 2 in the "Data collation and statistical analysis methods" section are from 2009 to 2017. Is the correct period? 

5 - This study provides the most extensive evidence regarding the trends in health management and health education on chronic diseases such as hypertension and diabetes in Inner Mongolia, China. What are the practical and policy implications of this study?

6 - Highlight the main limitations and suggestions for future research.

Author Response

We thank the reviewer for the insightful comments. Please see the attachment for point-by-point responses.

Reviewer 2 Report

It will be great to see the health management and health education by SES and how it changes over time. In addition, The references cited are thin with focus on studies in China. It will be helpful to cite some WHO documents on why chronic diseases management and education are important as theoretical background to support this study.  

Author Response

(The authors gave the same response as above.)

Reviewer 3 Report

This is a very informative paper on preventive education and disease management practice in a large region of China. The descriptive nature of the paper signifies the increased emphasis on health promotion and disease control efforts made in the Inner Mongolian region. The preventive benefits are demonstrated by the increasing trend of preventive and educational programs and practices.

Several conceptual and methodological issues pertaining to the study are noted as follows:

1. Conceptual Clarification on Standardized Preventive Education or Health Management Practice:  From the public health perspective, health promotion and disease control could be classified into primary, secondary, and tertiary prevention.  The paper needs to clarify the content of health management practice.  More specifically, the authors should define health management or disease management in the context of public health.  In the western world, health management refers to activities or programs centered on improving the effectiveness and efficiency of management practice of health organizations at the institutional level.  However, the preventive efforts of the public health sector are geared towards primary or secondary prevention. 

2. Expected Outcomes or Performance Metrics in Public Health Practice: What was the expectation for the preventive program to achieve on the annual basis?  Were there specific goals or performance levels described in the program design?

3. Sampling of Records:  It appears to me that the records were systematically sampled.  Did you randomly start picking up the records? What was the sampling frame?  Was a weighting procedure used?

4. Limitations:  Ideally, an experimental procedure (either a simple experimental design or a quasi-experimental design) should be performed for the evaluation of health promotion and disease prevention programs.

5.  Logic Model for Evaluating Preventive Practice or Disease Management:

The authors could employ a generic evaluation framework as the logic model used by public health researchers.

In summary, although this is an informative paper on public health practice and evaluation, the paper should carefully address the above issues as noted above before it is recommended for further consideration.

Author Response

(The authors gave the same response as above.)
